# Comparative Transcriptome Analysis of Two Sugarcane Cultivars in Response to Paclobutrazol Treatment

**DOI:** 10.3390/plants11182417

**Published:** 2022-09-16

**Authors:** Ronghua Zhang, Haibi Li, Yiyun Gui, Jinju Wei, Kai Zhu, Hui Zhou, Prakash Lakshmanan, Lianying Mao, Manman Lu, Junxian Liu, Youxiong Que, Song Li, Xihui Liu

**Affiliations:** 1Sugarcane Research Center, Chinese Academy of Agricultural Sciences, Nanning 530007, China; 2Guangxi Key Laboratory of Sugarcane Genetic Improvement, Guangxi Academy of Agricultural Sciences, Nanning 530007, China; 3Guangxi South Subtropical Agricultural Science Research Institute, Guangxi Academy of Agricultural Sciences, Longzhou 532415, China; 4National Engineering Research Center for Sugarcane, Fujian Agriculture and Forestry University, Fuzhou 350002, China

**Keywords:** comparative transcriptome analysis, paclobutrazol, RNA−seq, sugarcane

## Abstract

Sugarcane is an important crop across the globe, and the rapid multiplication of excellent cultivars is an important object of the sugarcane industry. As one of the plant growth regulators, paclobutrazol (PBZ) has been frequently used in the tissue culture of sugarcane seedlings. However, little is known about the molecular mechanisms of response to PBZ in this crop. Here, we performed a comparative transcriptome analysis between sensitive (LC05−136) and non−sensitive (GGZ001) sugarcane cultivars treated by PBZ at three time points (0 d, 10 d, and 30 d) using RNA sequencing (RNA−Seq). The results showed that approximately 70.36 Mb of clean data for each sample were generated and assembled into 239,212 unigenes. A total of 6108 and 4404 differentially expressed genes (DEGs) were identified within the sensitive and non−sensitive sugarcane cultivars, respectively. Among them, DEGs in LC05−136 were most significantly enriched in the photosynthesis and valine, leucine and isoleucine degradation pathways, while in GGZ001, DEGs associated with ion channels and plant–pathogen interaction were mainly observed. Notably, many interesting genes, including those encoding putative regulators, key components of photosynthesis, amino acids degradation and glutamatergic synapse, were identified, revealing their importance in the response of sugarcane to PBZ. Furthermore, the expressions of sixteen selected DEGs were tested by quantitative reverse transcription PCR (RT−qPCR), confirming the reliability of the RNA−seq data used in this study. These results provide valuable information regarding the transcriptome changes in sugarcane treated by PBZ and provide an insight into understanding the molecular mechanisms underlying the resistance to PBZ in sugarcane.

## 1. Introduction

Sugarcane (*Saccharum* spp.) is one of the largest crops in the world. To date, this crop has received great attention due to its huge commercial value in sugar and biofuel production [1]. Therefore, the development and improvement of sugarcane cultivars are essential for meeting the continuously increasing requirement for sugar and biofuel. How to increase rapid multiplication has long been a problem of great concern in sugarcane breeding programs. Micropropagation, which plays a vital role in the quick spread of new cultivars and superior crop cultivars, has been widely used for the rapid multiplication of sugarcane cultivars. Plant growth regulators (PGRs) are also known as intrinsic factors which play an essential role in plant tissue culture [2]. As one of the PGRs, paclobutrazol (PBZ) has been frequently used in tissue culture. Evidence has revealed that PBZ could inhibit the gibberellin biosynthesis and induce a variety of morphological and then physiological changes in plants, such as increasing the photosynthetic pigments [3,4], improving nutrient uptake [5,6], and enhancing flowering and seed yields [5,7]. Nevertheless, little is known about the molecular mechanisms of sugarcane response to PBZ treatment.

To improve the sugar yield and its adaption, several prevalent commercial sugarcane cultivars have been cultivated in China, such as LC05−136 and GGZ001. It is anticipated that the rapid spread of these superior cultivars through micropropagation will bring a huge economic value to the sugarcane industry. It is well known that the exogenous hormones including PGRs and cytokinin are needed at different breeding stages during tissue culture in sugarcane [8]. Sugarcane cultivars have different breeding coefficients, resulting in the various proportions of exogenous hormones separately or in combination in the culture medium. Previous research has demonstrated that PBZ has an inhibitory effect on the vegetative growth of sugarcane tissue−cultured seedlings but can promote the proliferation of tissue−cultured seedlings. For example, Daniels et al. [9] confirmed that the medium supplemented with 0.08% PBZ had the best effect for the multiplication of sugarcane cultivar CPCL99−4455. Liu et al. [10] reported that soaking the seed with PBZ (PP333) at a concentration of 50 mg/L could effectively improve the tillering of sugarcane seedling (ROC22). They also found that the highest content of chlorophyll and soluble protein in leaves was observed when the sugarcane seedling was treated with 90 mg/L PP333, while the proline content and peroxidase activity were the highest in sugarcane treated with 50 mg/L PP333 [10]. These results confirmed that PP333 could significantly increase the proliferation rate of tissue culture in sugarcane. However, there is still no report on the genetic basis of the growth difference in sugarcane cultivars treated by PBZ. Therefore, identifying the key genes, regulatory factors and networks involved in the regulation and proliferation is essential for revealing the molecular mechanism of the proliferation in sugarcane tissue−cultured seedlings, which is also of great significance to shorten the cultivation period and save costs.

RNAcSeq is a common and effective approach to monitor transcriptional changes, providing a comprehensive understanding of the underlying genes and their interaction networks mediated by biotic and abiotic factors [11,12,13]. It has been widely used to investigate the molecular mechanisms of sugarcane response to biotic and abiotic factors. For example, a comparative transcriptome analysis was conducted for resistant and sensitive sugarcane cultivars in response to infection by *Xanthomonas albilineans* [14], while Belesini et al. [15] investigated the transcriptome profiles between drought−tolerant and drought−sensitive sugarcane cultivars under multiple drought stress conditions. In the present study, de novo transcriptome in the two sugarcane cultivars (non−sensitive to PBZ: GGZ001 and sensitive to PBZ: LC05−136) treated by PBZ was performed, and then the molecular mechanisms of sugarcane in response to PBZ were analyzed. The present study aims to provide novel insights into the molecular mechanisms in sugarcane triggered by PBZ treatment, which should constitute useful gene resources for molecular breeding in sugarcane.

## 2. Results

### 2.1. Statistics of RNA−seq Data

Eighteen libraries were sequenced using the Illumina sequencing platform, generating 690.65 Mb for LC05—136 and 685.38 Mb for GGZ001, respectively (Table 1). After filtering, approximately 70.36 Mb clean data for each sample were generated, with an average mapping rate of 85.01%. A total of 239,212 unigenes were found, with a mean length of 1790 bp, GC percentage of 47.64%, and an N50 of 2541 bp (Table 2). Most unigenes had a length of >2000 bp, accounting for 35.71% of the total unigenes. The longest and shortest sequences were 15,177 bp and 297 bp, respectively.

### 2.2. Functional Annotation for Unigenes

The functions of all identified assembled unigenes were predicted using the BLASTx program against seven public databases: KEGG, GO, NR, NT, SwissProt, Pfam and KOG (Figure 1). The results showed that a total of 239,212 unigenes were identified from Nr 185,226 (77.43%), 195,891 from Nt (81.89%), 12,995 from SwissProt 5 (54.33%), 133,402 from KOG (55.77%), 143,333 from KEGG (59.92%), 140,499 from GO (58.73%), and 135,189 from Pfam (56.51%). Interestingly, all these unigenes could be annotated in at least one of the seven databases. Among them, 81,725 unigenes (34.16%) were present in all seven screened databases.

### 2.3. Differential Expression Analysis for Sensitive Cultivar LC05−136

With adjusted *p*−value < 0.01 and a fold change (FC) >2 based on the DESeq2 method, a total of 6108 unigenes were found to be differentially expressed between the sensitive cultivar LC05−136 treated by PBZ and control (Figure 2). Compared with the control, 1884 downregulated and 1060 upregulated DEGs were found at 10 d (Figure 2A). Meanwhile, 2855 downregulated and 1682 upregulated DEGs were detected at 30 d (Figure 2B). At 10 d and 30 d after treatment, 606 and 1228 DEGs were specifically upregulated in LC05−136 (Figure 2D), whereas 965 and 1936 DEGs were downregulated in LC05−136, respectively (Figure 2C). During the 10 d–30 d period, the downregulated DEGs in LC05−136 had a higher number than the upregulated ones (2901 vs. 1834).

### 2.4. Differential Expression Analysis for Non−Sensitive Cultivar GGZ001

Using the same threshold mentioned above, a total of 4404 DEGs were detected within the non−sensitive cultivar GGZ001 (Figure 3). We observed that 225 and 158 genes were downregulated and upregulated DEGs in GGZ001 at 10 d, respectively (Figure 3A). For the GGZ001 at 30 d, a total of 1745 downregulated and 2423 upregulated DEGs were observed (Figure 3B). After treatment, 126 downregulated and 110 upregulated DEGs were specifically expressed in GGZ001 at 10 d, respectively (Figure 3C,D). Meanwhile, 1646 downregulated and 2375 upregulated DEGs were determined in GGZ001 at 30 d, respectively (Figure 3C,D). During the 10 d–30 d period, the number of downregulated DEGs was lower than that of upregulated DEGs in GGZ001 (1772 vs. 2485).

### 2.5. Validation of DEGs within Sensitive and Non−Sensitive Sugarcane Cultivars

RT−qPCR (quantitative real time PCR) analysis was conducted for sixteen DEGs in LC05−136 and GGZ001 (Figure 4). The results showed that the expression levels of these DEGs in LC05−136 displayed a similar trend with that of the RNA−Seq (Figure 4A). In GGZ001, except for Unigene89351_ALL, the expression trend of these DEGs revealed by RT−qPCR was similar to that from the RNA−Seq data (Figure 4B). These results suggested that our RNA−Seq data were reliable and suitable for further transcriptome analysis.

### 2.6. GO Functional Analysis of DEGs within Sensitive and Non−Sensitive Sugarcane Cultivars

To explore the potential function of the DEGs, GO enrichment analysis was conducted in the two sugarcane cultivars. Venn analysis showed that a total of 373 DEGs were shared between LC05−136 and GGZ001 (Appendix A). They were significantly assigned into two main GO categories: “Biological process” and “Molecular function” (Appendix A). The most significantly enriched GO term of biological process and molecular function was the trehalose metabolic process and trehalose−phosphatase activity, respectively.

For the downregulated DEGs, a total of 16,828 DEGs were assigned into 277 GO terms in LC05−136 (Appendix A), while the 7244 DEGs in GGZ001 were attributed to 192 GO terms (Appendix A). For the upregulated DEGs, a total of 7358 DEGs were enriched into 169 GO terms in LC05−136 (Appendix A), while for those 8803 DEGs in GGZ001, they were assigned into 194 GO terms (Appendix A). The top five GO enrichment analysis of downregulated and upregulated DEGs is presented in Figure 5A,B, respectively, indicating the difference between LC05−136 and GGZ001. For the downregulated DEGs from LC05−136, the most significantly enriched terms of the biological process, cellular component, and molecular function were the “generation of precursor metabolites and energy”, “organelle subcompartment”, and “calcium ion binding”, respectively. For the downregulated DEGs from GGZ001, the most significant terms “hormone metabolic process”, “cell periphery”, and “transporter activity” were observed in the biological process, cellular component, and molecular function, respectively. Moreover, for the upregulated DEGs, “L−arabinose metabolic process” and “regulation of response to osmotic stress” were the most significant terms of the biological process in LC05−136 and GGZ001, respectively. The terms “cell periphery” and “nucleus” were the most representative of a cellular component in LC05−136 and GGZ001, respectively. The terms “heme binding” and “transcription regulator activity” were found to be the most representative of molecular function in LC05−136 and GGZ001, respectively.

### 2.7. KEGG Enrichment Analysis of DEGs within Sensitive and Non−Sensitive Sugarcane Cultivars

To further explore the biological pathway of DEGs in sugarcane triggered by PBZ, these DEGs were mapped onto the KEGG database using TBtools v1.051. In response to PBZ treatment, 5987 and 4403 DEGs were significantly enriched in the KEGG pathways in LC05−136 and GGZ001, respectively (Figure 6A). Interestingly, more downregulated DEGs (3564) in LC05−136 were activated, while more upregulated DEGs (2265) in GGZ001 were observed (Figure 6A). In LC05−136, 54 pathways were significantly enriched (*p* < 0.01) after PBZ treatment (Figure 6B). Most of the downregulated DEGs were significantly enriched for the top three pathways “Photosynthesis”, “Metabolism”, and “Photosynthesis−antenna proteins”, while the upregulated DEGs in LC05−136 were most significantly enriched in the pathways of “Valine, leucine and isoleucine degradation”, “beta−Alanine metabolism”, and “Amino acid metabolism”. In GGZ001, we observed that these DEGs were distributed into 38 KEGG pathways (Figure 6C). For the downregulated DEGs, the pathways of “Ion channels”, “Transporters”, and “Tryptophan metabolism” were the top three significant enrichment pathways, while the upregulated DEGs were most significantly enriched in the “Plant−pathogen interaction”, followed by the “Organismal Systems” and “Environmental adaptation”.

### 2.8. DEGs Involved in Metabolism for Sensitive Cultivar LC05−136

Considering that most of the DEGs were enriched into metabolic pathways, we speculated that they might play important roles in growth and development in LC05−136 treated by PBZ. In this study, the downregulated and upregulated DEGs associated with the most significant metabolism pathway were further explored in the LC05−136. As shown in Figure 7, a total of 29 downregulated DEGs were significantly enriched in the photosynthesis pathway. More detailed information on the downregulated DEGs is listed in Appendix A. Among them, 12 and 10 DEGs were aligned into photosysm II and photosysm I, respectively. Meanwhile, two downregulated DEGs were distributed into the cytoc hrome b6/f complex, photosynthethic electron transport, and F−type ATPase, respectively. Moreover, we observed that a total of 15 upregulated DEGs were significantly enriched in the pathway of “Valine, leucine and isoleucine degradation” (Appendix A). Detailed information on the 15 DEGs is listed in Appendix A.

### 2.9. DEGs Associated with Plant–Pathogen Interaction for Non−Sensitive Cultivar GGZ001

Considering that most DEGs in GGZ001 were enriched into the plant–pathogen interaction pathway, we focused on their roles in response to PBZ. We observed that 173 upregulated DEGs corresponding to 31 genes were most significantly enriched in the plant–pathogen interaction pathway (Figure 8). Detailed information on the 31 DEGs is listed in Appendix A. Interestingly, we found that 24 upregulated DEGs corresponding to four members (WRKY1, WRKY2, WRKY29, and WRKY33) of the WRKY transcription factor family might be related to various important physiological responses of the GGZ001. In addition, a total of 71 upregulated DEGs were involved in the function of calcium and calmodulin, followed by 25 upregulated DEGs associated with disease resistance protein, and 15 upregulated DEGs corresponding to two members (MPK3 and MEKK1) of the mitogen−activated protein kinase gene family involved in the transduction of external signals. Moreover, 27 downregulated DEGs corresponding to six genes were most significantly enriched in the ion channels. Interestingly, two genes (GRIK2 and GRIP) were involved in the glutamatergic synapse pathway. Detailed information on the identified downregulated DEGs is listed in Appendix A.

## 3. Discussion

RNA−seq is the most powerful and attractive tool for deeply investigating the transcriptional characteristics of sugarcane in response to biotic and abiotic stresses [15,16,17,18]. In this study, de novo assembly of two sugarcane cultivars was performed using the RNA−seq platform, generating approximately 70.36 Mb clean data for each sample and assembling into 239,212 unigenes. Among them, a total of 81,725 unigenes were present in all seven screened databases, including Nr, Nt, SwissProt, KEGG, Pfam, KOG, and GO. The high−quality data were then used to characterize and compare global gene expression profiles after PBZ treatment between two sugarcane cultivars. A total of 6108 and 4404 DEGs were found within the sensitive cultivar LC05−136 and non−sensitive cultivar GGZ001, respectively. The results suggested that the global gene expression within the 30 d after sugarcane was treated by PBZ was more intense in the sensitive than the non−sensitive cultivar. We also observed that the number of downregulated DEGs (3820) was higher than the upregulated DEGs (2288) in LC05−136, in contrast, a higher number of upregulated DEGs (2533) were found in GGZ001 compared to the downregulated DEGs (1871). This indicated that gene downregulation was more active in the sensitive cultivar, while gene upregulation appeared to be predominant in the non−sensitive cultivar.

To obtain a comprehensive understanding of the biological function of DEGs, GO enrichment analysis was first conducted. In the present study, our data showed that 373 common DEGs in the two sugarcane cultivars were most significantly enriched in the trehalose metabolic process of biological process and trehalose−phosphatase activity of molecular function, respectively. Trehalose plays a vital role in the regulation of plant metabolism and development [19]. Ibrahim and Abdellatif [20] further found that the trehalose had a significant and positive effect on most growth parameters and foliar applications with trehalose−induced water stress tolerance in wheat plants. They suggested that maybe these significantly enriched common DEGs might be related to the response of sugarcane to PBZ treatment. For the downregulated DEGs, we observed that the most significantly enriched DEGs for LC05−136 and GGZ001 were activated in the molecular function terms “calcium ion binding” and “transporter activity”, respectively. For the upregulated DEGs, the terms “heme binding” and “transcription regulator activity” were the most representative of molecular function in LC05−136 and GGZ001, respectively. Huang et al. [21] found that PBZ mainly regulated the dwarfism mechanism of Hippeastrum through affecting various biological processes in plants, and pathway enrichment analysis showed that differential genes were significantly enriched in metabolic pathways and biosynthesis of secondary metabolites. The above results suggested that these DEGs in sensitive (LC05−136) and non−sensitive (GGZ001) sugarcane cultivars might respond to PBZ treatment through different biological processes.

KEGG analysis revealed that a total of 54 and 38 pathways were significantly (*p* < 0.01) enriched in LC05−136 and GGZ001 after PBZ treatment, respectively. For the sensitive cultivar LC05−136, the downregulated and upregulated DEGs were most significantly enriched in the metabolism pathway. It is noted that a total of 29 downregulated DEGs were significantly enriched in the photosynthesis pathway. We observed that 12 and 10 DEGs were aligned into photosysm II and photosysm I, respectively. Meanwhile, two downregulated DEGs were distributed into the Cytochrome b6/f complex, Photosynthethic electron transport, and F−type ATPase, respectively. It is well known that photosynthesis is one of the major functions that drives plant growth and development [22]. Rossato et al. [23] found that the photosynthesis rate was negatively affected by both borer and spittlebug infestations, resulting in yield losses of sugarcane. PBZ can significantly affect the photosynthetic capacity of plants by changing the morphological and physiological changes of plant leaves, and these changes include the reduction of leaf area [24,25] and the increase of chlorophyll content [26,27,28]. The results suggested that these downregulated DEGs might inhibit the photosynthesis of LC05−136 treated by PBZ, thereby influencing the growth of sugarcane seedling. Moreover, a total of 15 upregulated DEGs were enriched in the valine, leucine and isoleucine degradation pathway, which indicated that these genes might promote the degradation of valine, leucine and isoleucine of LC05−136 treated by PBZ, which was in accordance with a previous study that various amino acids play an important role in the shoot regeneration of sugarcane [29]. For the non−sensitive cultivar GGZ001, 173 upregulated DEGs corresponding to 31 genes were most significantly enriched in the plant–pathogen interaction pathway. Among them, we found that 24 upregulated DEGs corresponding to four members (*WRKY1*, *WRKY2*, *WRKY29*, and *WRKY33*) of the WRKY transcription factor family might be related to various important physiological responses of the non−sensitive cultivar GGZ001. A large number of studies have demonstrated that the WRKY family in the plant has an important biological function response to different kinds of biotic and abiotic stresses and working mechanisms [30,31,32]. Interestingly, in our study, 15 upregulated DEGs corresponding to two members (*MPK3* and *MEKK1*) of the mitogen−activated protein kinase (MAPK) gene family might be involved in the transduction of external signals. Notably, phosphorylation of a WRKY transcription factor by MAPKs plays a critical role in plant response to pathogens and stress conditions [33]. For example, Ali et al. [34] reported that the members of MAPK cascade gene families regulate adverse stress responses through multiple signal transduction pathways in sugarcane. Other studies have shown that PBZ can improve the resistance level of plants to abiotic stress by affecting light cooperation [26,27,28]. Therefore, we have every reason to believe that the interaction between WRKY and MAPK plays an essential role in the adaption of the non−sensitive cultivar GGZ001 in response to PBZ. In addition, a total of 71 upregulated DEGs were involved in the function of calcium and calmodulin, followed by 25 upregulated DEGs associated with disease resistance protein. In plants, the calmodulin−regulated proteins and disease resistance protein play vital roles in the abiotic stress responses [35] and disease resistance [36], respectively. Our results suggested that WRKY family genes might be related to stress responses in sugarcane treated by PBZ.

## 4. Materials and Methods

### 4.1. Plant Materials and RNA Isolation

Two sugarcane cultivars (non−sensitive to PBZ: GGZ001 and sensitive to PBZ: LC05−136) from the Sugarcane Research Institute, Guangxi Academy of Agricultural Sciences were used in the present study. Samples of roots were collected for RNA−sequencing from the two selected cultivars at 0, 10 d, and 30 d. Sampling for each timepoint was performed in triplicate. These samples were immediately snap−frozen in liquid nitrogen and stored at −80 °C until use. The total RNA of each sample was isolated using a TRIzol™ Reagent (Thermo Fisher Scientific, Wilmington, NC, USA) following the manufacturer’s instructions. The quality of RNA was evaluated using a NanoDrop 2000 spectrophotometer (Thermo Fisher Scientific, Wilmington, NC, USA) and an Agilent 2100 bioanalyzer (Agilent Technologies, Santa Clara, CA, USA).

### 4.2. cDNA Library Construction and Sequencing

A total of 18 cDNA libraries were successfully constructed in this study. In brief, the total RNA for each sample was first treated and isolated with DNaseI and oligo−dT beads, respectively, aiming to obtain the ploy(A) mRNAs. Subsequently, these mRNAs were reverse transcripted into the first−strand cDNA using the reverse transcriptase, and the second−strand cDNA was synthesized using DNA polymerase I and RNaseH. Next, the cDNA was ligated with an adaptor or index adaptor using T4 DNA ligase. Finally, the adaptor−ligated fragments were separated by the agarose gel electrophoresis, and then the cDNA fragments were amplified using the PCR. The libraries were sequenced using an Illumina HiSeq 25,000 at BGI company (BGI, Shenzhen, China). The Illumina sequencing data of sugarcane treated with PBZ were deposited into the National Center for Biotechnology Information (NCBI) SRA database under accession number PRJNA719604.

### 4.3. De Novo Assembly and Functional Annotation

The raw data for each sample were checked using the TrimGalore v0.6.6 [37] software through the removal of adaptor sequences, reads with ambiguous sequences “N”, and low−quality reads, aiming to generate clean data. The clean data were de novo assembled into the sugarcane transcriptome using the Trinity v2.11.0 [38] software. The non−redundant unigenes, as long as possible, were obtained through the sequence splicing and redundancy removal from all sample unigenes. Only assembled unigenes with lengths of >200 bp were included in subsequent analyses.

To annotate the unigenes, these sequences were aligned and annotated to the following databases: NCBI non−redundant protein sequences (Nr; https://www.ncbi.nlm.nih.gov/guide/, accessed on 25 August 2021), NCBI non−redundant nucleotide sequences (Nt; https://www.ncbi.nlm.nih.gov/guide/, accessed on 25 August 2021), protein family (Pfam; http://www.pfam.org/, accessed on 25 August 2021), gene ontologies (GO; http://geneontology.org/, accessed on 25 August 2021), KEGG Orthology database (KO; https://www.genome.jp/kegg/ko.html/, accessed on 25 August 2021), a manually annotated and reviewed protein sequence database (Swiss−Prot; https://web.expasy.org/docs/swiss−prot_guideline.html/, accessed on 25 August 2021), and Clusters of Orthologous Groups of proteins (KOG/COG; ftp://ftp.ncbi.nih.gov/pub/COG/KOG/, accessed on 25 August 2021), with an E−value cut−off of 1 × 10^−5^.

### 4.4. Analysis of Differentially Expressed Genes (DEGs)

The differential expression analysis of the pairwise comparison groups was performed using the DESeq2 v3.11 [39] software after obtaining the counts of genes. The transcripts per million (TPM) values for each gene were calculated. Genes with an adjusted *p*−value < 0.05 and an absolute value of log2ratio (treatment/control) ≥ 1 were considered as the differentially expressed genes (DEGs). GO and KEGG enrichment analysis of the DEGs were performed using the TBtools v1.051 [40] software. The threshold of adjusted *p*−value < 0.05 and 0.001 was used for the GO and KEGG enrichment analysis, respectively.

### 4.5. RT−qPCR Analysis

First−strand cDNA synthesis was performed using the HiScript II Q RT SuperMix for qPCR (+gDNA wiper) (Vazyme Biotech Co., Ltd., Nanjing, China) following the manufacturer’s instructions. RT−qPCR was performed using the ChamQ Universal SYBR qPCR Master Mix (Vazyme Biotech Co., Ltd., Nanjing, China). Each reaction was performed in triplicate. The relative mRNA levels were calculated using the 2^−ΔΔCt^ method and normalized by the *GAPDH* gene as the internal reference gene. The primers of 16 unigenes were designed using Primer5.0 software and listed in Appendix A.

## 5. Conclusions

In summary, we compared and characterized the transcriptome data between sensitive and non−sensitive sugarcane cultivars in response to PBZ treatment. We identified a total of 6108 and 4404 DEGs within LC05−136 and GGZ001, respectively. We confirmed several key pathways or candidate genes involved in the response of two sugarcane cultivars to PBZ treatment. Our findings provide valuable information on the transcriptome changes in sugarcane treated by PBZ and provide novel insights into the understanding of the molecular mechanisms underlying the response to PBZ treatment in sugarcane.

## Figures and Tables

**Figure 1 plants-11-02417-f001:**
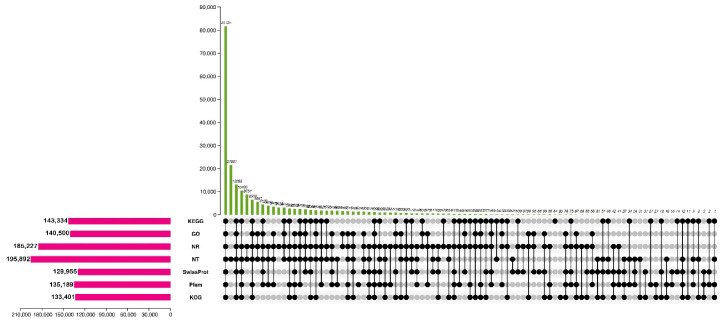
BLAST hits for unigenes against seven public databases.

**Figure 2 plants-11-02417-f002:**
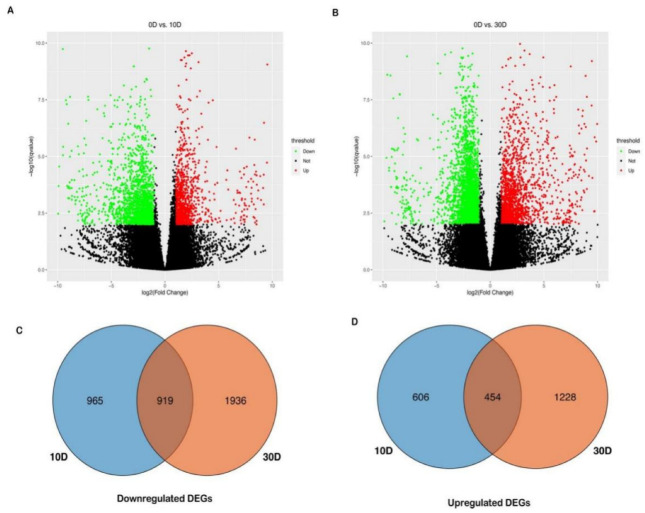
Differential gene expression in sensitive cultivar LC05−136 treated by PBZ at 10 d and 30 d. (**A**) Differential expression analysis of samples between 0 d and 10 d; (**B**) Differential expression analysis of samples between 0 d and 30 d; (**C**) Venn analysis of downregulated DEGs between 10 d and 30 d; (**D**) Venn analysis of upregulated DEGs between 10 d and 30 d.

**Figure 3 plants-11-02417-f003:**
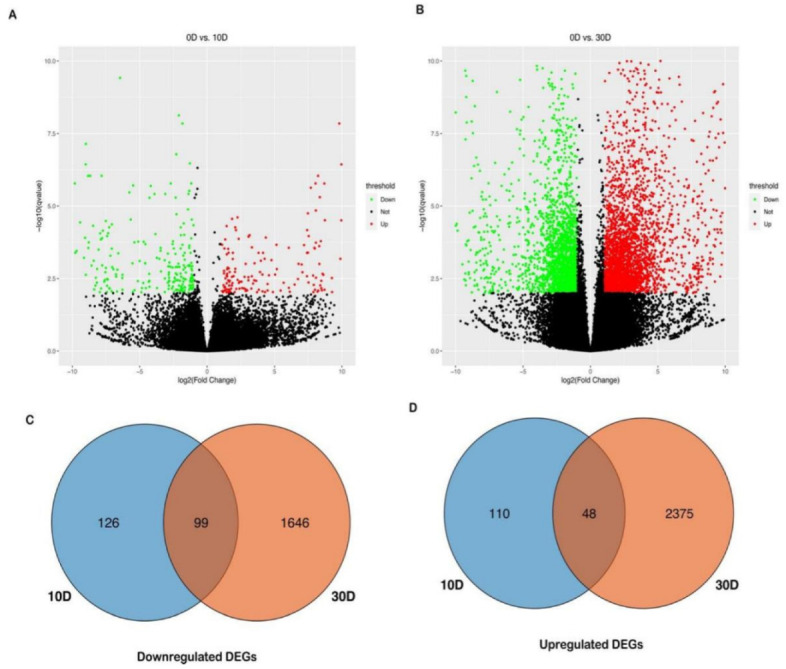
Differentially expression analysis of non−sensitive cultivar GGZ001 treated by PBZ at 10 D and 30 D. (**A**) Differential expression analysis of samples between 0 d and 10 d; (**B**) Differential expression analysis of samples between 0 d and 30 d; (**C**) Venn analysis of downregulated DEGs between 10 d and 30 d; (**D**) Venn analysis of upregulated DEGs between 10 d and 30 d.

**Figure 4 plants-11-02417-f004:**
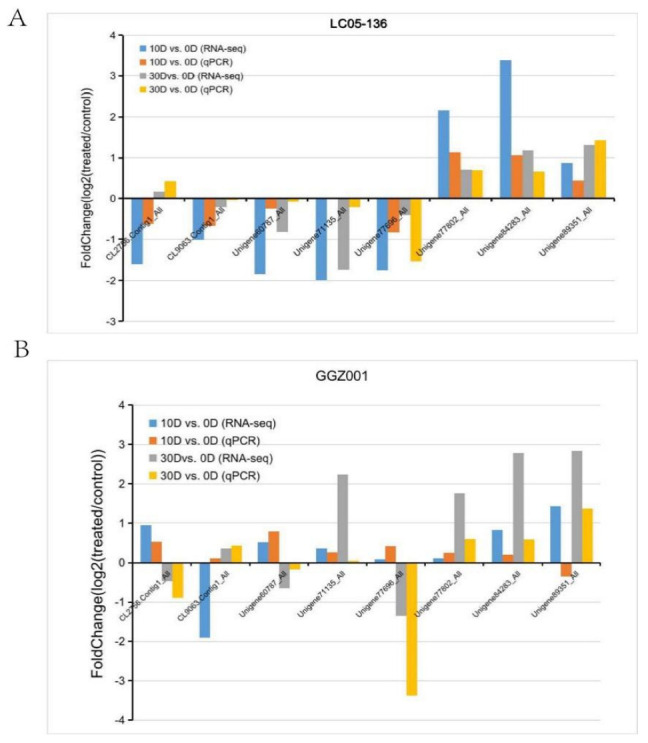
Relative expressions of representative transcripts in LC05−136 (**A**) and GGZ001 (**B**) using RT−qPCR.

**Figure 5 plants-11-02417-f005:**
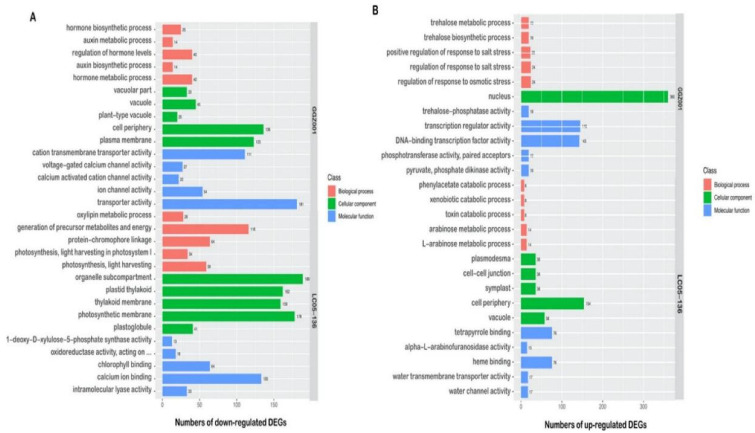
GO enrichment analysis of downregulated (**A**) and upregulated (**B**) DEGs in LC05−136 and GGZ001.

**Figure 6 plants-11-02417-f006:**
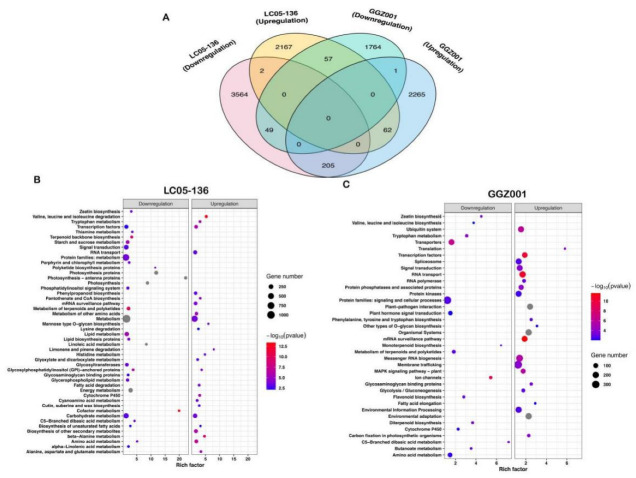
KEGG pathway classification and functional enrichment of DEGs. (**A**) Venn diagrams of DEGs assigned to KEGG pathways in two sugarcane cultivars treated by PBZ. The upregulated DEGs of LC05−136 and GGZ001 are shown in yellow and blue, respectively; the downregulated DEGs of LC05−136 and GGZ001 are shown in pink and green, respectively. (**B**) Pathway functional enrichment of up− and downregulated DEGs in LC05−136. (**C**) Pathway functional enrichment of up− and downregulated DEGs in GGZ001.

**Figure 7 plants-11-02417-f007:**
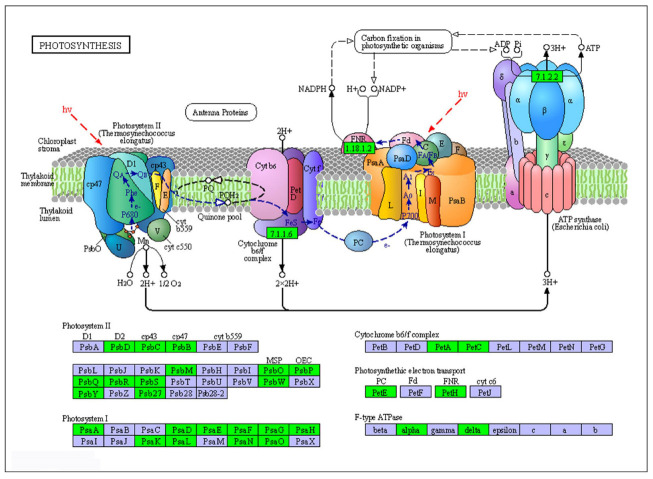
The photosynthesis pathways and the expression profile of DEGs in LC05−136. The green color represents the downregulated DEGs.

**Figure 8 plants-11-02417-f008:**
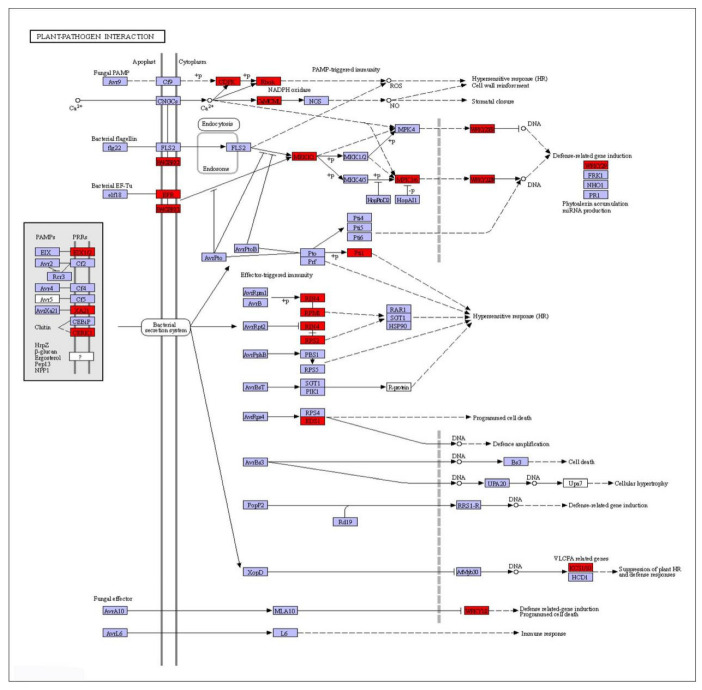
The photosynthesis pathways and expression profile of DEGs in GGZ001. The red color represents the upregulated DEGs.

**Table 1 plants-11-02417-t001:** Sequence statistics of the sugarcane transcriptome.

Sample	Total Raw Reads (M)	Total Clean Reads (M)	Total Mapping (%)	Clean Reads Q20 (%)	Clean Reads Q30 (%)	Clean Reads Ratio (%)
LC05−136−10D−1	77.13	70.7	82.92	95.87	87.09	91.67
LC05−136−10D−2	77.13	70.85	84.43	95.72	86.67	91.86
LC05−136−10D−3	77.13	70.85	84.11	95.64	86.48	91.87
LC05−136−30D−1	77.13	71.13	84.29	96.28	87.76	92.22
LC05−136−30D−2	77.13	70.91	84.04	96.36	87.91	91.94
LC05−136−30D−3	77.13	71.34	84.13	96.53	88.37	92.5
LC05−136−0D−1	75.37	69.59	84.17	96.18	87.55	92.33
LC05−136−0D−2	75.37	69.16	83.97	96.12	87.33	91.75
LC05−136−0D−3	77.13	71.19	84.25	96.25	87.76	92.31
GGZ001−0D−1	73.62	68.55	86.25	96.42	88.31	93.12
GGZ001−0D−2	77.13	70.82	85.36	96.11	87.42	91.83
GGZ001−0D−3	75.37	69.71	86.72	96.11	87.39	92.49
GGZ001−10D−1	77.13	71.04	85.72	95.72	86.56	92.11
GGZ001−10D−2	75.37	68.5	85.95	95.56	86.21	90.89
GGZ001−10D−3	75.37	68.78	85.31	95.62	86.34	91.26
GGZ001−30D−1	77.13	71.27	86.21	95.97	86.89	92.4
GGZ001−30D−2	77.13	71.22	86.47	96.11	87.3	92.35
GGZ001−30D−3	77.13	70.82	85.88	96.07	87.22	91.82

**Table 2 plants-11-02417-t002:** Length distribution of assembled unigenes.

Unigenes	Number	Percentage (%)
200–500 bp length	43,548	18.20
500–1000 bp length	39,173	16.38
1000–2000 bp length	71,079	29.71
>2000 bp length	85,412	35.71
Total	239,212	100%
Minimum length (bp)	297	/
Mean length (bp)	1790	/
Maximum length (bp)	15,177	/
N50	2541	/
N90	999	
GC%	47.64	/

## Data Availability

The data presented in this study are available within the article and its Appendix A.

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
