# Peer review of "Comparative Transcriptome Analysis of Two Sugarcane Cultivars in Response to Paclobutrazol Treatment"

_plants, 2022, doi:10.3390/plants11182417_

Round 1

Reviewer 1 Report

My comments are below:

 Abstract 

Abstract is carelessly written authors should incorporate their notable findings and adequately connect with the sentences they choose to correspond.

Introduction

  • The introduction section must have a clear hypothesis and significantly develop the second paragraph of your manuscript. Make it more connecting to the problem statement. 
  • Overall there is the repetition of the information, which could be avoided.

Discussion 

  • This section should include more information and references related to the relevant and related works. 

Conclusions

  • If possible, restructure and carefully edit the conclusion section and add clear information regarding the most noteworthy findings.

The introduction is relatively poor.

The discussion is poor as well. 

Reviewer 2 Report

In this study, authors conducted transcriptome analyses of two sugarcane cultivars in response to PBZ treatment.

I think the title should be changed as follows.

Comparative transcriptome analysis of two sugarcane cultivars in response to paclobutrazol treatment

L40 Please include scientific name of sugarcane.

L69 Reported -> reported

Introduction is nicely written. It was easy to read and understand the background of this study.

I did not still understand about resistance and susceptible sugarcane cultivars against PBZ treatment. Please explain this in introduction. In my opinion, authors should use other expressions instead of resistance and susceptible cultivars.

2.1. RNA sequencing and Assembly -> Please provide subtitle in detail.

Most subtitles did not provide information for the results. They are too short. Please rewrite them.

In addition, I don’t understand why authors conducted de novo transcriptome assembly although there is a reference genome for sugarcane.

In the beginning of the result section, authors should describe the treatment of PBZ on the sugarcane and the results for total RNA extraction, library preparation, and RNA sequencing. Moreover, Table S1 should be presented in the main manuscript. Otherwise, it was very hard to understand the materials used for RNA sequencing.

Authors should provide high quality figures. Most figures were very poor quality.

I found that there were too many DEGs in this study. Authors should reanalyze their data associated with DEGs using read counts (mapped read number on each gene). For DEG analysis using DESeq, read counts should be used. However, authors used TPM values as follows. In addition, TPM values can not be calculated by DESeq2 program.

L385-388 Please revise this paragraph.

For the counts of each library, DESeq2 v3.11 [31] software was used to estimate the transcripts per million (TPM) values for each gene. The differential expression analysis of the two groups was also performed using the DESeq2 v3.11 [31] software.

All related results associated with DEGs should be rewritten after conducting DEG analysis using read counts.

Data analyses for DEG were totally wrong. Therefore, this manuscript should be rejected now. However, I encourage to resubmit their manuscript after revision.

Round 2

Reviewer 1 Report

The manuscript has improved significantly. Therefore, it can be accepted for publication.   

Reviewer 2 Report

Authors properly revised their manuscript according to reviewer's comments. 

I suggest this manuscript for publication as it is.